# Non-specific effects of Pneumococcal and Haemophilus vaccines in children aged 5 years and under: a systematic review

Keith Geraghty [1], Darragh Rooney,[1] Chris Watson,[2] Mark T Ledwidge,[3] Liam Glynn,[1] Joe Gallagher [4]

¹School of Medicine, University of Limerick, Limerick, Ireland
²Wellcome-Wolfson Institute for Experimental Medicine, Queen's University Belfast, Belfast, UK
³Health Research Institute, University College Dublin College of Health Sciences, Dun Laoghaire, Ireland
⁴Global Health, Irish College of General Practitioners, Dublin, Ireland

**Correspondence to**
Dr Keith Geraghty;
gerokeith@gmail.com

## ABSTRACT

**Objective** To determine the evidence for non-specific effects of the Pneumococcal and Haemophilus influenza vaccine in children aged 5 years and under.

**Data sources** A key word literature search of MEDLINE, EMBASE, The Cochrane Central Register of Controlled Trials, the European Union Clinical Trials Register and ClinicalTrials.gov up to June 2023.

**Study eligibility criteria** Randomised controlled trials (RCTs), quasi-RCT or cohort studies.

**Participants** Children aged 5 or under.

**Study appraisal and synthesis methods** Studies were independently screened by two reviewers, with a third where disagreement arose. Risk of bias assessment was performed by one reviewer and confirmed by a second. Results were tabulated and a narrative description performed.

**Results** Four articles were identified and included in this review. We found a reduction in hospitalisations from influenza A (44%), pulmonary tuberculosis (42%), metapneumovirus (45%), parainfluenza virus type 1–3 (44%), along with reductions in mortality associated with pneumococcal vaccine. No data on the Haemophilus vaccine was found.

**Conclusions and implications** In this systematic review, we demonstrate that there is a reduction in particular viral infections in children aged 5 years and under who received the 9-valent pneumococcal conjugate vaccine which differ from those for which the vaccine was designed to protect against. While limited studies have demonstrated a reduction in infections other than those which the vaccine was designed to protect against, substantial clinical trials are required to solidify these findings.

**PROSPERO registration number** CRD42020146640.

### STRENGTHS AND LIMITATIONS OF THIS STUDY

⇒ This systematic review explored the potential non-specific effects of two commonly administered vaccines in children aged 5 and under.

⇒ This systematic review was based on the systematic literature search following Preferred Reporting Items for Systematic review and Meta-Analysis.

⇒ Three of the articles reviewed were a post hoc analysis of a phase III randomised, double-blind placebo-controlled 9-valent pneumococcal conjugate vaccine trial, using the same data set.

⇒ No studies were found which explored the Haemophilus vaccine in isolation.

## INTRODUCTION

The beneficial effects of vaccination in reducing morbidity and mortality from the illnesses they are designed to prevent is widely reported with estimates that vaccines prevent approximately 6 million deaths worldwide annually.[1][2] Vaccination has attempted to reduce morbidity and mortality from major diseases such as diphtheria, tetanus, yellow fever, pertussis, Haemophilus influenza type b disease, measles, mumps, rubella, typhoid and rabies, with the successful eradication of smallpox and near-complete eradication of poliomyelitis.[3]

Pneumonia remains a major cause of morbidity and mortality worldwide. The WHO estimates there to be 156 million cases of pneumonia each year in children younger than 5 years, with as many as 20 million requiring hospital admission, accounting for an estimated 0.935 million deaths every year.[4][5] However, the number of pneumonia deaths in young children has almost halved between 2000 and 2015.[6] This is likely attributable to improved provision of primary care through the Integrated Management of Childhood Illness programme, and increasing use of universal vaccination, including Haemophilus influenza type B (Hib) and pneumococcal vaccines.[7]

Vaccination with the Hib and pneumococcal conjugate vaccines protect children from invasive disease caused by these organisms. Routine vaccination has been shown to dramatically decrease the incidence of invasive Hib disease in children leading to a reduction in the number of hospital

admissions.[8 9] A study by Cowgill *et al*[9] reported a reduction of the incidence of invasive Hib disease to 12% of its baseline level 3 years after the vaccine was introduced. The 7-valent pneumococcal conjugate vaccine (PCV7) has been shown to reduce incidence rates of invasive pneumococcal disease by as much as 75%.[10] The introduction of the 13-valent pneumococcal conjugate vaccine (PCV13), replacing PCV7 in 2010, is likely to further reduce the incidence of pneumonia due to enhanced coverage of serotypes responsible for the majority of pneumococcal pneumonia cases in children worldwide.[11] GAVI, the Vaccine Alliance—previously the Global Alliance for Vaccines and Immunisation—has been pivotal in the introduction of vaccination protocols enabling an estimated reduction of between 6 and 7.5 million cases of pneumonia and the avoidance of between 230 000 and 290 000 deaths since its introduction of its pneumococcal vaccine programme in 2007.[12]

Furthermore, there is a body of evidence that certain vaccines yield beneficial effects not just limited to the illnesses they are designed to prevent. These effects are commonly termed 'non-specific' effects. Vaccination with measles-containing vaccines, Bacille Calmette-Guérin (BCG) and oral polio vaccine (OPV) have all been shown to reduce both hospitalisation rates and mortality in children with further research required to better understand the impact on mortality.[13–15] Non-specific beneficial effects of vaccination in pneumonia may be due to bacteria and viruses acting as co-pathogens in the aetiology of pneumonia.[16] Evidence also indicates that respiratory viruses contribute to bacterial infections, often leading to bacterial superinfections.[16] Non-specific beneficial effects of vaccination may be due to the results of altered immune system memory, leading to a reduction in all-cause mortality.[17 18] One study proposes that live vaccines can induce innate immune training, producing pro-inflammatory responses to unrelated antigens.[19] However, there is contradictory evidence, with one systematic review demonstrating that receipt of the diphtheria pertussis tetanus (DPT) vaccine was shown to be associated with an increase in all-cause mortality. This should be interpreted with caution as all 10 studies within the analysis were observational and classed as having a 'high risk of bias'.[19 20]

In recent years, researchers have offered a compelling argument for a radical change in the current vaccine paradigm,[21] and newly suggested principles could mean a drastic overhaul in how countries administer their vaccines programmes. For example, it is suggested that live vaccines yield non-specific effects which are beneficial,[22–26] while non-live vaccines may have detrimental non-specific effects, especially for women.[27 28] In addition, it is proposed that non-specific effects of vaccines are determined by their most recent vaccine.[29]

Further work is required to determine the presence or absence of non-specific effects of newer vaccines such as pneumococcus and Haemophilus influenza which were more recently added to schedules internationally. In this study, we aim to determine if the administration of pneumococcal and/or Hib vaccines in infancy is associated with an effect on survival or hospitalisations from infections other than those conditions that the vaccine is designed to prevent in children up to 5 years of age. Furthermore, we aim to explore if the administration of pneumococcal and/or Hib vaccines in infancy are associated with an effect on severity of the illness as defined by the authors, or hospitalisations for infections other than those conditions that the vaccine is designed to prevent in children up to 5 years of age.

## METHODOLOGY

The systematic review protocol was registered with the PROSPERO database, which guided the authors in conducting this review. The Preferred Reporting Items for Systematic Reviews and Meta-Analyses statement[30] was used to guide the reporting and conduct of the review.

### Eligibility criteria

To be eligible for inclusion, studies had to meet certain predetermined criteria outlined in the review protocol. The population of interest was children aged 5 and under. Haemophilus vaccines along with any formulation of the pneumococcal vaccine, including PCV7, PCV13 and PCV23 were included. Studies reporting survival, all-cause mortality, or deaths from infections other than those conditions that the vaccine was designed to prevent, or studies reporting deaths from all causes (eg, all-cause mortality, child survival) or severity as defined by the authors (eg, hospitalisation) were eligible for inclusion. Only randomised controlled trials (RCTs), quasi-RCTs or cohort studies written in English were included in this review.

### Exclusion criteria

Ecological studies, uncontrolled studies (ie, case reports and case series studies), 'case-only' studies and self-controlled case series studies were excluded from this review, as these studies provide less reliable data for assessing non-specific effects of vaccine on hospitalisations and mortality. Additionally, animal and laboratory studies were excluded. Studies which included adults or children over the age of 5, or those which did not report children under the age of 5 separately, were not included in this review.

### Data management

Results from the literature search were directly imported into the online Rayyan software package.[31] This was used for screening through the citations and abstracts from the electronic databases, as well as the full-text articles indicated via the search. Two reviewers independently screened the literature search results for inclusion. They then independently reviewed the full text of potentially relevant articles and screened them to determine inclusion

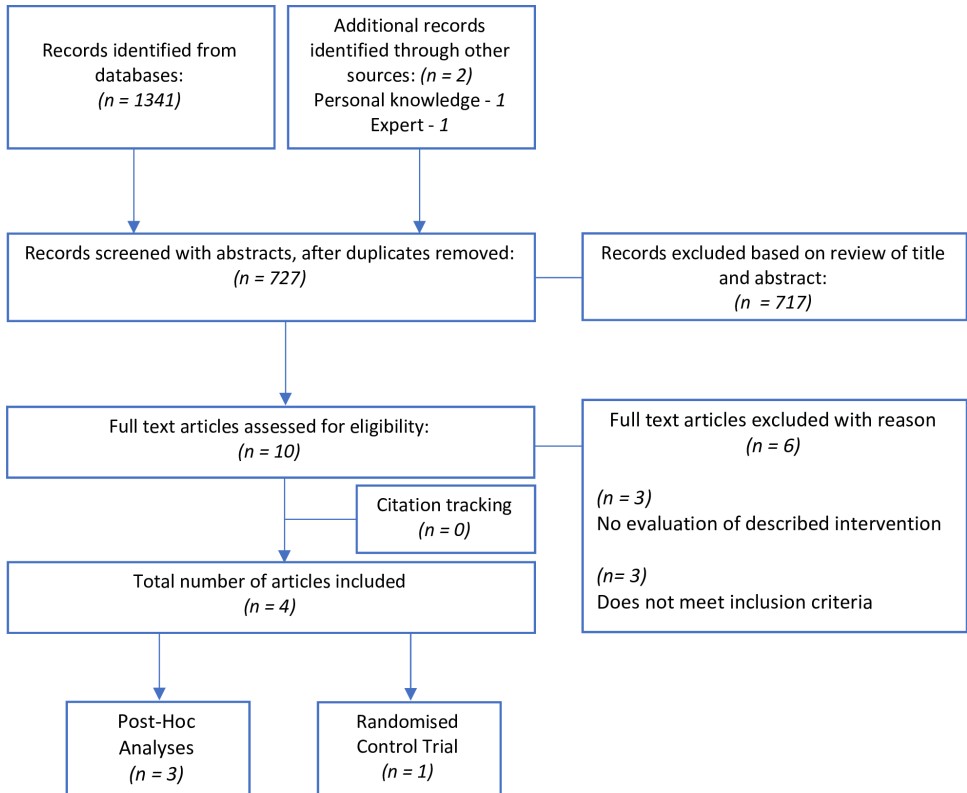

**Figure 1** Preferred Reporting Items for Systematic Reviews and Meta-Analyses flow chart.

using the same inclusion and exclusion criteria. In cases of disagreement a third reviewer acted as arbitrator.

### Search strategy

The search strategy was jointly developed and tested during the protocol phase and was subsequently finalised by reviewers prior to commencing the search. Medical subject headings and text words related to children, pneumococcal and Haemophilus influenza vaccination were used. Electronic databases MEDLINE (Ovid interface, 1948 onwards),[32] EMBASE (Ovid interface, 1980 onwards)[33] and PubMed were each searched using the predetermined search terms. The Cochrane Central Register of Controlled Trials (Wiley interface, current issue)[34] and European Union Clinical Trials Register (EURACT) were searched to supplement the search. These databases were last searched end of June 2023. All reference lists of included studies were scrutinised and reviewed to further identify relevant reviews not found during the database search. Finally, a bibliography of included articles was circulated to experts in the field to ensure literature saturation. The flow chart of the study identification process is illustrated in figure 1.

### Data collection process

To ensure data accuracy, two reviewers independently abstracted the data using a standardised data abstraction form. This form was developed during the protocol phase and was trialled and refined prior to commencement of the review.

### Risk of bias assessment

Two researchers independently assessed risk of bias using the Cochrane Collaboration Risk of Bias tool.[35] Guided by the recommendations of the Cochrane handbook, seven methodological items were used to assess each individual study's risk of bias, including randomised sequence generation, elective reporting, incomplete outcome data, blinding of outcome assessment, blinding of participants and personnel, method of concealing allocation and other potential biases. Risk of bias was subsequently classified as 'low risk, 'unclear' or 'high risk.' Discussion and consultation with a third reviewer took place in the case of disagreement between the two primary assessors.

### Data synthesis

To effectively synthesise the data obtained, data was grouped together and tabulated. Textual descriptions were provided. Studies were contrasted, and a narrative on reported outcomes, effectiveness of studies and design strengths and weaknesses were presented.

### Patient and public involvement

Patients or the public were not involved in the design, or conduct, or reporting, or dissemination plans of our research.

### Ethical approval

This review does not require ethical approval as authors have retrieved and synthesised previously published literature.

**Table 1** Study characteristics

| Authors | Study type | Country | Population | Subpopulation | Vaccine | Control | Other vaccines concurrently administered |
|---|---|---|---|---|---|---|---|
| Moore et al[36] | Post hoc analysis of a phase III randomised, double-blind placebo-controlled PCV9 trial. | South Africa | 39 836 children, aged 84 days and under. | HIV-infected and HIV-uninfected. | 9-valent pneumococcal vaccine or a placebo (2 µg of capsular polysaccharide (serotypes 1, 4, 5, 9V, 14, 19F and 23F), 4 µg of serotype 6B and 2 µg of oligosaccharide 18C). | Haemophilus influenzae type b conjugate vaccine. | Diphtheria, tetanus and whole-cell pertussis (DTwP, Aventis Pasteur); hepatitis B (Hepaccine-B, Cheil Sugar Organisation); and oral live, trivalent poliovirus types 1, 2 and 3 (Poli-oral, Biovac). |
| Madhi et al[37] | Post hoc analysis of a phase III randomised, double-blind placebo-controlled PCV9 trial. | South Africa | 39 836 children, aged 84 days and under. | HIV-infected and HIV-uninfected. | 9-valent pneumococcal vaccine or a placebo (2 µg of capsular polysaccharide (serotypes 1, 4, 5, 9V, 14, 19F and 23F), 4 µg of serotype 6B and 2 µg of oligosaccharide 18C). | Haemophilus influenzae type b conjugate vaccine. | Diphtheria, tetanus and whole-cell pertussis (DTwP, Aventis Pasteur); hepatitis B (Hepaccine-B, Cheil Sugar Organisation); and oral live, trivalent poliovirus types 1, 2 and 3 (Poli-oral, Biovac). |
| Madhi et al[38] | Post hoc analysis of a phase III randomised, double-blind placebo-controlled trial. | South Africa | 39 836 children (aged 60–84 days). | HIV-infected and HIV-uninfected. | 9-valent pneumococcal vaccine or a placebo (2 µg of capsular polysaccharide (serotypes 1, 4, 5, 9V, 14, 19F and 23F), 4 µg of serotype 6B and 2 µg of oligosaccharide 18C). | Haemophilus influenzae type b conjugate vaccine. | Diphtheria, tetanus and whole-cell pertussis (DTwP, Aventis Pasteur); hepatitis B (Hepaccine-B, Cheil Sugar Organisation); and oral live, trivalent poliovirus types 1, 2 and 3 (Poli-oral, Biovac). |
| Dagan et al[40] | Randomised, double-blind placebo-controlled PCV9 trial. | Israel | 264 children (aged 12–35 months). | - | 9-valent pneumococcal conjugate vaccine (2 µg of capsular polysaccharide (serotypes 1, 4, 5, 9V, 14, 18C, 19F and 23F), 4 µg of serotype 6B, coupled to diphtheria toxin CRM197). | Meningococcus group C conjugate vaccine. | Not reported. |

PCV9, 9-valent pneumococcal conjugate vaccine.

 Geraghty K, et al. BMJ Open 2023;13:e077717. doi:10.1136/bmjopen-2023-077717

**Table 2** Results

| Study | Subpopulation | Illness | Outcome measured | Efficacy (95% CI) | P value |
|---|---|---|---|---|---|
| Moore et al[36] | HIV-uninfected | First episode culture-confirmed PTB | All-cause hospitalisations | 35.3 (–38.1 to 69.7) | 0.2562 |
| | | All categories of first episode PTB | | 0.0 (–40.9 to 29.1) | 0.9983 |
| | | All culture-confirmed PTB | | 38.9 (–29.3 to 71.1) | 0.1931 |
| | HIV-infected | First episode culture-confirmed PTB | | 47.3 (8.6 to 69.6) | 0.0203 |
| | | All categories of first episode PTB | | 13.2 (–10.4 to 31.8) | 0.2475 |
| | | All culture-confirmed PTB | | 43.6 (5.5 to 66.4) | 0.0274 |
| | Overall | First episode culture-confirmed PTB | | 43.4 (11.5 to 63.8) | 0.0114 |
| | | All categories of first episode PTB | | 8.8 (–11.7 to 25.5) | 0.9983 |
| | | All culture-confirmed PTB | | 42.1 (11.2 to 62.3) | 0.0113 |
| Madhi et al[37] | HIV-uninfected | Any virus-associated pneumonia | All-cause hospitalisations | 33 (15 to 48) | 0.0008 |
| | | Influenza A | | 34 (–14 to 62) | 0.1 |
| | | RSV | | 32 (6 to 50) | 0.02 |
| | | PIV types 1–3 | | 41 (–10 to 68) | 0.09 |
| | | Adenovirus | | 31 (–62 to 70) | 0.4 |
| | HIV-infected | Any virus-associated pneumonia | | 0.2 (–14 to 47) | 23 |
| | | Influenza A | | 57 (7 to 80) | 0.03 |
| | | RSV | | –30 (–140 to 31) | 0.4 |
| | | PIV types 1–3 | | 50 (–17 to 78) | 0.1 |
| | | Adenovirus | | –150 (–1188 to 51) | 0.3 |
| | Overall | Any virus-associated pneumonia | | 31 (15 to 43) | 0.0004 |
| | | Influenza A | | 45 (14 to 64) | 0.01 |
| | | RSV | | 22 (–3 to 41) | 0.08 |
| | | PIV types 1–3 | | 44 (8 to 66) | 0.02 |
| | | Adenovirus | | 7 (–94 to 55) | 0.9 |
| Madhi et al[38] | HIV-uninfected | **Human metapneumovirus-associated LRTI** | All-cause hospitalisations | | |
| | | <6.0 months | | 6 (–83 to 51) | 0.87 |
| | | 6.1–12 months | | 52 (7 to 75) | 0.027 |
| | | 12.1–24 months | | 33 (–17 to 62) | 0.16 |
| | | >24 months | | 47 (–13 to 76) | 0.09 |
| | HIV-infected | <6.0 months | | 34 (–295 to 89) | 0.65 |
| | | 6.1–12 months | | 43 (–93 to 83) | 0.36 |
| | | 12.1–24 months | | 58 (–64 to 89) | 0.20 |
| | | >24 months | | 42 (–46 to 77) | 0.24 |
| | Overall | <6.0 months | | 14 (–59 to 53) | 0.63 |
| | | 6.1–12 months | | 52 (14 to 73) | 0.012 |
| | | 12.1–24 months | | 40 (–1 to 64) | 0.053 |
| | | >24 months | | 45 (1 to 70) | 0.04 |
| Dagan et al[40] | – | **URI** | Hospitalisations | | |
| | | <36 months | | 17 (0.7 to 0.99) | 0.036 |
| | | >36 months | | 14 (0.73 to 1.02) | 0.082 |
| | | Overall | | 15 (0.76 to 0.96) | 0.009 |

Continued

**Table 2**  Continued

| Study | Subpopulation | Illness | Outcome measured | Efficacy (95% CI) | P value |
|---|---|---|---|---|---|
| | | **All-cause LRTI** | | | |
| | | <36 months | | 23 (0.62 to 0.95) | 0.015 |
| | | >36 months | | 9 (0.73 to 1.12) | 0.371 |
| | | Overall | | 16 (0.72 to 0.98) | 0.024 |
| | | **Otitis media** | | | |
| | | <36 months | | 23 (0.58 to 1.03) | 0.075 |
| | | >36 months | | 12 (0.65 to 1.20) | 0.425 |
| | | Overall | | 17 (0.67 to 1.02) | 0.078 |

LRTI, lower respiratory tract infection; PIV, parainfluenza virus; PTB, pulmonary tuberculosis; RSV, respiratory syncytial virus; URI, upper respiratory infection.

## RESULTS

We retrieved 1341 articles from MEDLINE, EMBASE and PubMed electronic databases using our search strategy. Rayyan software automatically removed duplicate records, and suggested articles which were similar in title or author, allowing manual removal of any duplicates not successfully detected by the software package. We had prior knowledge of one article which met our inclusion criteria, which was therefore included in the screening process. One further article was suggested by an expert in the field, and this was also included. After duplicates were removed, 727 records remained and were subsequently screened by two reviewers independently. Of these, 717 were excluded based on screening abstracts and titles. Ten full-text articles were assessed for eligibility, using the eligibility form. Six of these were excluded as they did not meet inclusion criteria (n=3), or they did not evaluate the described intervention (n=3). Scanning references from the remaining articles did not reveal any new articles. Therefore, four studies were included in our systematic review. Three studies were post hoc analyses of the same RCT data set. One study was a randomised control study.

All four studies were associated with the use of the 9-valent pneumococcal conjugate vaccine (PCV9). No study involving the use of the Haemophilus influenza vaccine was identified.

Considering the use of common data sets, 40 100 children under the age of 5 were involved in these studies. Summaries of the main findings of each study are outlined in table 1. Results of these studies are summarised in table 2.

Three of the studies,[36–38] all of which undertook a post hoc analysis of a prospective randomised, double-blind placebo-controlled PCV9 efficacy study demonstrated a reduction in hospitalisations from infections other than those that the PCV was designed to prevent against. The trial, conducted in South Africa, enrolled 39 836 children between 1 March 1998 and October 2000. Participants received either PCV9 or a control, and investigators and laboratory staff remained blinded to the randomisation throughout the surveillance phase. The mortality rate was reduced by 5% among all children (p=0.58), and by 6% among HIV-infected children (p=0.63) in this study.

On analysis of this data set, Moore et al[36] determined that there was a reduction in hospitalisation for culture-confirmed pulmonary tuberculosis (PTB) of 43.4% in the vaccine study group versus the control group (95% CI: 11.5 to 63.8; p=0.0114). In HIV-infected children, the risk of being hospitalised with a lower respiratory tract infection was 47.3% lower (95% CI: 8.6% to 69.6%; p=0.0203) in PCV9 recipients compared with the placebo recipients. Although not statistically significant, a similar trend was observed in children uninfected by HIV (relative risk reduction 35.3% (95% CI: −38.1% to 69.7%); p=0.2562). Where available, intention-to-treat results are described as it reflects the practical clinical scenario and gives an unbiased estimate of treatment effect.[39]

In their analysis of the same data set, Madhi et al[37] show that in all children (ie, those with or without HIV infection), PCV9 reduced pneumonias associated with any of the identified viruses by 31% (95% CI: 15% to 43%; p=0.0004). Furthermore, there were similar point estimates of efficacy with influenza A (45%; 95% CI: 14% to 64%; p=0.01), parainfluenza virus type 1–3 (44%; 95% CI: 3% to 64%; p=0.01) and respiratory syncytial virus (22%; 95% CI: −3% to 41%; p=0.08) in all children.

In a further study, Madhi et al[38] demonstrated a reduction in incidence of human metapneumovirus in both HIV-infected and HIV-unaffected children who received the PCV9 vaccine, (reduction of 53% (95% CI: 3% to 77%; p=0.035) vs 45% reduction (95% CI: 19% to 62%; p=0.002), in the per-protocol analysis, respectively). In this study, the authors also note a significant reduction in the incidence of clinical pneumonia among vaccine recipients overall (58; 95% CI 34 to 73; p=0.0002).

Dagan et al[40] conducted a small randomised, double-blind, placebo-controlled study in Israel, whereby 264 toddlers received either PCV9, or a control (meningococcal conjugate vaccine). This study revealed a 23% reduction (95% CI: 0.62% to 0.95%; p=0.015) in all-cause lower respiratory infection in children under 36 months compared with placebo recipients. A 17% reduction in

upper respiratory tract infection was observed in the same age group (95% CI: 0.7 to 0.99; p=0.036). Although there was a similar trend in children >36 months, this was not statistically significant.

We used the Cochrane tool for assessing risk of bias in clinical trials[35] for randomised studies. All studies were judged based on predetermined criteria, as discussed in the Methods sections of this paper. The studies included in this review were deemed low risk of bias. Furthermore, we considered any potential confounders and based on the location of the trials, found the socioeconomic status of the population to be a potential confounder for both the South African and Israeli studies. Certain populations are more likely to live in overcrowded communities, potentially subjecting them to a higher burden of disease.[41] Furthermore, we felt socioeconomic status was important as it may determine inequalities in healthcare access.[41] In all papers, there was no discussion which raised possible implications for vaccinating children in areas where disease may be more prevalent.

## DISCUSSION

Reports to date have determined the efficacy of PCV9 in the reduction of nasopharyngeal pneumococcal carriage,[42] invasive infection[43] and mucosal infections.[43] In this systematic review, we demonstrate that in some cohorts, there is a reduction in particular viral infections in children under the age of 5 who received the PCV9 vaccine, which differ from those for which the vaccine was designed to protect against.

Reduction in the incidence of hospitalisation from other diseases was reported most frequently. Post hoc analyses from a phase III RCT in South Africa reveal a reduction in culture-confirmed PTB,[36] influenza A,[37] parainfluenza type 1–3[37] and human metapneumovirus[38] in all children, irrespective of their HIV status. In a smaller RCT in Israel,[40] a reduction in all-cause lower respiratory tract infections was determined in daycare centre attendees.

The authors of the phase III RCT included analysis of HIV-infected versus non-infected children, which allowed post hoc analysis of these subgroups. While the hospital reductions mentioned above include all children, the reduction of PTB in vaccine recipients is only statistically significant in HIV-infected children and overall.[36] Furthermore, the reduction of respiratory syncytial virus (RSV) was only significant in all-children, and those not infected with HIV.[37] All-cause mortality did not differ between groups in this trial.

While included trials have a relatively equal representation of both sexes in their population, there is no analysis of potential non-specific effects of the respective vaccines on men versus women. As described, several non-live vaccines, namely DPT,[44] inactive polio vaccine,[45] hepatitis B vaccine,[46] measles and yellow fever vaccine[47] are associated with increased female mortality. Given the pneumococcal vaccine is inactivated, further work exploring the sex-differential is warranted.

Interestingly, the control used in the RCT in South Africa was the Hib conjugate vaccine, the second vaccine we sought to explore in this review. This was given so those children in the control group could infer some benefit. It is therefore likely that these subsets would receive some protection against Haemophilus influenza. No study was identified which sought to explore non-specific effects in the Hib vaccine in isolation, and further research is required in this domain.

The South African trial reported on other vaccinations children received as part of their routine schedule, although did not outline the sequence in which these vaccines were administered. Unfortunately, this information was omitted from the Dagan study. Certain combinations of vaccines are associated with increased mortality (eg, DTP and measles vaccine),[48] whereas others are associated with lower mortality (eg, DTP and OPV).[49] In addition, there is evidence that combining non-live and live vaccines can have a detrimental effect on mortality.[50] Therefore, further detail regarding the sequence of vaccines is crucial, as it may impact efficacy and mortality.

This study has several limitations. Most notably, three of the articles reviewed were a post hoc analysis of the same data set, which used the same population. Post hoc analyses typically describe findings for which the parent study was not designed to explore. In addition, the high proportion of studies in this area which come from common data sets is a weakness of this review. As discussed, analysis to include sex-differential and sequencing of vaccines is missing from these studies.

The findings of this review do however add to the body of evidence that certain vaccines infer additional non-specific effects. The most widely reported of these vaccines are the BCG and OPV,[14 15] with some currently hypothesising that the BCG vaccine may in fact offer protection against the SARS-CoV-2 virus.[51]

Vaccination programmes have demonstrated wide-ranging benefits, however some of these are poorly defined. For public health officials and those deciding on health policy, there is a need to prioritise vaccines which have an identifiable and preventable disease burden. Consideration of those non-specific effects which may impact on morbidity and mortality are important in deciding on novel vaccination strategies.

## CONCLUSION

This review highlights the paucity of literature exploring the non-specific effects of the PCV and Haemophilus vaccine. While limited studies have demonstrated a reduction in infections other than those which the vaccine was designed to protect against, substantial clinical trials are required to solidify these findings.

**Contributors** KG—acquisition of data; analysis and interpretation of data; drafting of manuscript, primary reviewer. DR—conception and design of study; protocol design and registration; third reviewer. CW—reviewer of manuscript. MTL—reviewer of manuscript. LG—reviewer of manuscript. JG—conception and design

of study; protocol design and registration; critical revisions of manuscript; primary reviewer. KG is responsible for the overall content as the guarantor.

**Funding** This study was conducted with the financial support of the Bill and Melinda Gates Foundation (OPP113957) and Science Foundation Ireland (SFI)/ Department of Foreign Affairs (DFA) under the SDG Challenge Grant Number SFI/21/ FIP/SDG/9948.

**Competing interests** None declared.

**Patient and public involvement** Patients and/or the public were not involved in the design, or conduct, or reporting, or dissemination plans of this research.

**Patient consent for publication** Not applicable.

**Ethics approval** Not required.

**Provenance and peer review** Not commissioned; externally peer reviewed.

**Data availability statement** Data are available upon reasonable request.

**ORCID iDs**
Keith Geraghty http://orcid.org/0000-0002-1679-4445
Joe Gallagher http://orcid.org/0000-0002-5564-2890

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
