## [Reviewer comments · BMJ Open]

ARTICLE DETAILS

TITLE (PROVISIONAL)	A Systematic Review of Non-Specific Effects of Pneumococcal and Haemophilus Vaccines in Children Aged Five Years and Under.
AUTHORS	Geraghty, Keith; Rooney, Darragh; Watson, Chris; Ledwidge, Mark; Glynn, Liam; Gallagher, Joe

VERSION 1 – REVIEW

REVIEWER	Christian, Wieg Klinikum Aschaffenburg-Alzenau, Department of Neonatology, Klinikum Aschaffenburg, Aschaffenburg, Germany.
REVIEW RETURNED	20-May-2022

GENERAL COMMENTS	The manuscript is well written meeting the aim of the study expressed in the last sentence of the introduction . The study design including the search methods and the study selection is well chosen . The limitations of the presented study are clearly defined. As the discussion is written precisely meeting the main result of the study, I miss a short comment on the potential causes leading th the beneficial side effects on prevention of viral infections. There is only one hint in the introduction, but it may be worth to suggest further aspects to study by discussing the hypothetic underlying mechanisms.
---

REVIEWER	Sørup, Signe Aarhus University Hospital, Department of Clinical Epidemiology
REVIEW RETURNED	30-May-2022

GENERAL COMMENTS	Overall, the paper has an interesting subject and appear well written. However, there are some major issues as described below: Within the area of non-specific effect of vaccines it is generally hypothesised that live vaccines have beneficial effects and non-live vaccines might have detrimental non-speccific effects. Furthermore, it has often been seen that non-specific effects of vaccines are determined by the most recent vaccine (see e.g. https://pubmed.ncbi.nlm.nih.gov/32645296/). This perspective is not considered by the authors. It is important to include such considerations throughout the paper, in the introduction, in results (please describe any control vaccines used and what vaccines children might have gotten as part of the normal vaccination schedule), and the discussion. There is no aim or objective at the end of the introduction. However, the study questions are included in the methods section (I recommend to move these). The study questions specify that the following outcomes are of interest: " survival from infections other
--

	than those conditions that the vaccine is designed to prevent” and ” severity of the illness as defined by the authors or hospitalisations for infections other than those conditions that the vaccine is designed to prevent”. However, in the method section the following eligibility criteria are specified: ” Studies reporting survival, all-cause mortality, or deaths from infections other than those conditions that the vaccine was designed to prevent, and studies reporting deaths from all causes (e.g. all-cause mortality, child survival) were eligible”. The eligibility criteria do not match the study question completely particularly the eligibility lack the possibility that studies with outcomes of ” severity of the illness as defined by the authors or hospitalisations for infections other than those conditions that the vaccine is designed to prevent” can be reported. For this reason, I fear that not all relevant studies have been included or maybe the reporting of the study is not correct. This need to clarified. Table 1 are missing one article and only appear to include statistically significant results. All relevant results should be included. Please report clearly on all subpopulations with results. Please include information on age and potential other vaccines. For the results of reference 30 there appear to be some mistakes in the confidence limits. Why do you regard socioeconomic status as a confounder in randomised trials?
--	---

VERSION 1 – AUTHOR RESPONSE

Reviewer Comments	Author Response
Peer Reviewer 1 Prof. Wieg Christian, Klinikum Aschaffenburg-Alzenau	
“I miss a short comment on the potential causes leading to the beneficial side effects on prevention of viral infections. There is only one hint in the introduction, but it may be worth to suggest further aspects to study by discussing the hypothetic underlying mechanisms.”	Further hypotheses are referenced in the introduction which touch on the immunological processes that may underpin some of the non-specific effects discussed in the article.
Peer Reviewer 2 Dr. Signe Sørup, Aarhus University Hospital	
“Within the area of non-specific effect of vaccines it is generally hypothesised that live vaccines have beneficial effects and non-live vaccines might have detrimental non-specific effects. Furthermore, it has often been seen that non-specific effects of	A major literature review was undertaken by our team to explore these important hypotheses. Relevant papers were reviewed, and several incorporated into the introduction, results and discussion. We paid particular focus to sex-differential of non-specific effects, live-versus-

vaccines are determined by the most recent vaccine” – not considered by author..	non-live vaccines, and indeed, the sequence of vaccine administration.
“Please describe any control vaccines used and what vaccines children might have gotten as part of the normal vaccination schedule.”	Included in results, and in Table 1.
“There is no aim or objective at the end of the introduction. However, the study questions are included in the methods section (I recommend moving these)”	Moved as suggested.
“The eligibility criteria do not match the study question completely.”	Clarification of eligibility criteria – updated in Methodology.
“Table 1 are missing one article and only appear to include statistically significant results. All relevant results should be included. Please report clearly on all subpopulations with results. Please include information on age and potential other vaccines.”	We had a major revision to our table in this study. Two tables are now included in this piece of work. Table 1 highlights the important characteristics of each study such as population, vaccine in question, and control used. Table 2 provides detailed results of the studies.
“For the results of reference 30 there appear to be some mistakes in the confidence limits”	Amended.
“Why do you regard socioeconomic status as a confounder in randomised trials?”	A comment on this has been made in our Results section (page 18)